# Successful Postnatal Cardiopulmonary Resuscitation Due to Defibrillation

**DOI:** 10.3390/children8050421

**Published:** 2021-05-20

**Authors:** Lukas Peter Mileder, Nicholas Mark Morris, Stefan Kurath-Koller, Jasmin Pansy, Gerhard Pichler, Mirjam Pocivalnik, Bernhard Schwaberger, Ante Burmas, Berndt Urlesberger

**Affiliations:** 1Division of Neonatology, Department of Pediatrics and Adolescent Medicine, Medical University of Graz, 8036 Graz, Austria; nicholas.morris@medunigraz.at (N.M.M.); jasmin.pansy@medunigraz.at (J.P.); gerhard.pichler@medunigraz.at (G.P.); bernhard.schwaberger@medunigraz.at (B.S.); berndt.urlesberger@medunigraz.at (B.U.); 2Division of Pediatric Cardiology, Department of Pediatrics and Adolescent Medicine, Medical University of Graz, 8036 Graz, Austria; stefan.kurath@medunigraz.at (S.K.-K.); ante.burmas@medunigraz.at (A.B.); 3Pediatric Intensive Care Unit, Department of Pediatrics and Adolescent Medicine, Medical University of Graz, 8036 Graz, Austria; mirjam.pocivalnik@medunigraz.at

**Keywords:** neonate, resuscitation, ventricular fibrillation, defibrillation

## Abstract

An asphyxiated term neonate required postnatal resuscitation. After six minutes of cardio-pulmonary resuscitation (CPR) and two doses of epinephrine, spontaneous circulation returned, but was shortly followed by ventricular fibrillation. CPR and administration of magnesium, calcium gluconate, and sodium bicarbonate did not improve the neonate’s condition. A counter shock of five Joule was delivered and the cardiac rhythm immediately converted to sinus rhythm. The neonate was transferred to the neonatal intensive care unit and received post-resuscitation care. Due to prolonged QTc and subsequently suspected long-QT syndrome propranolol treatment was initiated. The neonate was discharged home on day 14 without neurological sequelae.

## 1. Introduction

Postnatal cardiac arrest is most commonly a consequence of failure in transition from placental to pulmonary gas exchange and hence secondary to a disturbance in establishing sufficient aeration of the lungs. Cardiac arrest is defined as the cessation of blood circulation resulting from absent or ineffective cardiac mechanical activity, which in neonates is primarily due to asystole, severe bradycardia (<60 beats per minute (bpm)) or pulseless electrical activity [1]. In the case of postnatal cardiac arrest despite effective ventilation, resuscitation guidelines recommend cardio-pulmonary resuscitation (CPR) with a ratio of three chest compressions to one ventilation [2]. Although intravenous epinephrine is rarely required during resuscitation in the delivery room [3], resuscitation guidelines recommend its use “if the heart rate has not increased to 60/min or greater after optimizing ventilation and chest compressions …” [2].

While defibrillation is one of the most effective interventions in the case of sudden cardiac arrest due to ventricular fibrillation (VF) or pulseless ventricular tachycardia in the adolescent and adult population, defibrillation is not mentioned in the neonatal resuscitation guidelines [2], and to our knowledge is yet to be reported in postnatal resuscitation.

## 2. Case Presentation

A male neonate was delivered at term (40^6/7^ weeks of gestation, birth weight 3400 g) by emergency caesarean section due to persistent bradycardia after premature rupture of membranes with clear amniotic fluid. He initially presented with reduced muscle tone, cyanosis, and a heart rate between 60 and 100 bpm. The neonate did not respond to drying and tactile stimulation, and had only insufficient breathing efforts. Electrocardiogram (ECG) and pulse oximetry monitoring were initiated and continuous positive airway pressure via face mask was applied, rapidly followed by non-invasive positive pressure ventilation (PPV) due to persistent bradycardia. Non-invasive PPV did not result in visible thoracic excursions and the peak inspiratory pressure was slowly increased from 25 cm H_2_O to 40 cm H_2_O during the second minute of life, but not leading to the expected improvement in the neonate’s condition. The attending neonatologist intubated the neonate, but no chest wall movement could be observed under invasive PPV und the neonate remained bradycardic around 40 bpm. CPR was commenced, but after doubting correct tube placement and failure to detect exhaled carbon dioxide (CO_2_) with a colorimetric CO_2_ detector the endotracheal tube was removed and CPR was continued under non-invasive PPV. The concentration of inspired oxygen was set to 1.0 and non-invasive PPV was delivered with a peak inspiratory pressure of 40 cm H_2_O, which now resulted in adequate thoracic excursions. In minute 9 after birth, intraosseous vascular access had been established and 30 µg epinephrine (~9 µg/kg) and 5 mL isotonic fluid were administered due to persistent bradycardia. However, incorrect needle placement was suspected leading to a second intraosseous puncture on the contralateral tibia. This time correct needle placement could be confirmed and a second dose of epinephrine (100 µg, ~29 µg/kg), followed by 5 mL of isotonic fluid, were successfully administered. This led to a rapid increase in heart rate to above 100 bpm and chest compressions were stopped after a total of six minutes of CPR. The airway was cleared of viscous secretions, the neonate was rapidly re-intubated under vision and the correct tube position was verified clinically and by detection of CO_2_ exhalation.

At 12 minutes after birth, approximately two minutes after the second dose of epinephrine and 90 s after return of spontaneous circulation, the ECG suddenly converted to VF with a heart rate of 280–340 bpm. During this episode the neonate deteriorated rapidly with clinical signs of shock, leading to a restart of CPR and the administration of 5 mL magnesium gluconate, 5 mL calcium gluconate 10%, and 5 mL sodium bicarbonate via the intraosseous access. After two minutes of CPR a defibrillator was available and self-adhesive pediatric defibrillation pads were placed in anterior-posterior position. A non-synchronized counter shock of five Joule was delivered causing an immediate conversion to sinus rhythm at a rate of 150–160 bpm with sufficient cardiac output. The neonate was transferred to the neonatal intensive care unit where standardized therapeutic hypothermia (72 h) was initiated because of hypoxic ischemic encephalopathy after perinatal asphyxia (Apgar 2/1/0). Postnatal resuscitation is summarized in Figure 1.

Before and during whole body cooling, we observed a prolonged QT interval corrected for heart rate (QTc) using Bazett’s formula of 0.6 s without any further ECG abnormalities. In response to the prolonged QTc, we initiated beta-blocker therapy with propranolol (2 mg/kg/day), and over the following days QTc decreased to stable values of 0.41 s. Complete echocardiographic evaluation was performed and did not show any structural abnormalities or congenital heart disease. Electrolytes were checked regularly during and after therapeutic hypothermia, and were all within normal ranges.

In the following, neither clinical signs nor laboratory findings made suggestion of any infection related issue. The neonate was extubated on day seven and in the comprehensive work-up including neurological examination, cranial sonography, electroencephalography, and magnetic resonance imaging no abnormalities were detected. After a further week of uneventful cardiac monitoring, the neonate was discharged home on day 14 of life with cardio-respiratory home monitoring and under continued propranolol treatment.

Genetic analysis revealed a mutation in the SCN5A gene (c.3911C>T, p.Thr1304Met). Extended genetic analyses found the identical gene mutation in the neonate’s brother and mother, both of whom were asymptomatic with normal QTc. There were no cases of sudden cardiac death reported in the extended family.

At the age of 18 months, neurodevelopmental testing was unremarkable and showed normal development. At the latest cardiology follow-up at the age of 3.5 years, the boy was asymptomatic with normal QTc without anti-arrhythmic medication.

## 3. Discussion

Several aspects of this unusual case are worth discussing and have the potential to improve both knowledge and management of hemodynamically compromised neonates during postnatal resuscitation.

To our knowledge, this is the first reported case of a neonate who required defibrillation after birth. During infancy, VF is generally a rare event, with an incidence of only 0.52 per 100,000 person-years during the first year of life [4]. During the neonatal period VF may be caused either by long-QT syndrome (LQTS) [5] or an anomalous left coronary artery descending from the pulmonary artery [6], first of which is a known cause of sudden infant death syndrome [7].

In our case the neonate developed VF shortly after the administration of epinephrine. This appears to have triggered VF on the basis of preexisting QTc prolongation, which is known to increase susceptibility to arrhythmogenic factors. Epinephrine shortens the effective refractory time of the atria, atrioventricular (AV) node and ventricular myocardium, improves conduction via the AV node, and, therefore, may induce sustained ventricular arrhythmia [8]. The electrophysiological effects of epinephrine mainly result from stimulation of beta-receptors and while it also stimulates alpha-receptors, this seems not to affect the AV node. However, alpha-receptor stimulation increases the effective refractory time of the atria and ventricles, partially offsetting the shortening of refractory time mediated by beta-receptor stimulation [8]. Epinephrine administration itself has been associated with prolongation of QTc and induction of Torsades de Pointes [9]. Byrum et al. [10] also described VF in a neonate with Wolff-Parkinson-White syndrome associated with the use of digitalis, which exhibits positive bathmotropic properties similar to epinephrine. In addition, ischemia and reperfusion during postnatal cardiac arrest may have contributed to cardiac arrhythmia [11]. High inspiratory oxygen supplementation may have resulted in hyperoxemia, which then may have increased the susceptibility for VF [12]. Finally, therapeutic hypothermia may have induced or further aggravated QTc prolongation [13], as Vega et al. [14] showed (reversible) prolonged QTc in neonates undergoing therapeutic hypothermia due to hypoxic ischemic encephalopathy.

Due to initially prolonged QTc, empirical propranolol treatment was established. In addition, we found a mutation in the SCN5A gene, which is present in 5–10% of patients with LQTS [15]. The SCN5A gene encodes the α-subunit of the cardiac sodium channel Nav1.5, which initiates and transmits action potentials within the myocardium, and gain-of-function mutations in SCN5A may cause LQTS [16]. In untreated patients with LQTS, cardiac arrest or sudden cardiac death is the sentinel event in 13% of cases [17]. However, in our patient the SCN5A mutation discovered is not associated with LQTS. The fact that our patient’s QTc normalized over time and that two family members carrying the same mutation were asymptomatic without QTc prolongation led us to exclude LQTS as an underlying condition, allowing us to discontinue beta-blocker treatment during follow-up. Furthermore, we had calculated QTc using Bazett’s formula, which is known to overestimate QTc at high heart rates [18]. Applying different formulas for QT correction (e.g., Fridericia, Framingham, Hodges) may yield more accurate QTc in newborns especially with heart rates above 100 bpm [18].

Case reports on defibrillation in neonates include an incidence of fire ignited by a defibrillation attempt in a 10-day-old neonate with VF following heart surgery [19] and the need for electrical cardioversion during the first day of life in one preterm and one term neonate with narrow complex tachycardia [20]. Sauer et al. [5] described defibrillation in a 19-day-old neonate due to VF, which was associated with susceptibility to LQTS type 6. Hirakubo et al. [21] reported on a term neonate suffering from tuberous sclerosis with multiple cardiac tumors and (supra-)ventricular tachycardia, who developed VF requiring electrical cardioversion on day 12 after birth. However, postnatal resuscitation requiring defibrillation due to VF has not been reported yet. It has to be noted that the energy dose used in our patient (five Joule, i.e., ~1.5 Joule per kg) was lower than the recommended initial dose of two to four Joule per kg for infants or children suffering from VF [22]. Despite reports about the potential inadequacy of initial shock doses of two Joule per kg for termination of VF or pulseless ventricular tachycardia in infants and children [23], defibrillation was successful on the first attempt in our patient. We speculate that due to the short plasma half-life ofepinephrine electrophysiological effects on the myocardium may have already been wearing off and that in this vulnerable state of arrhythmia a relatively low energy dose was sufficient. Nevertheless, we still regard the recommended energy dose of two to four Joule per kg for defibrillation of VF advisable [22].

From a clinical viewpoint, the severely compromised neonate was resuscitated according to published guidelines [1]. Immediate application of ECG leads allowed for rapid and continuous assessment of heart rate as well as early recognition of VF. This underlines the recommendation to use ECG monitoring during neonatal resuscitation [1,2]. Non-invasive PPV was initially not effective despite an increased peak inspiratory pressure, which at least in apneic preterm neonates is a common finding due to temporary airway obstruction [24]. The lack of exhaled CO_2_ following the first intubation attempt we consider most likely due to a displaced endotracheal tube causing esophageal intubation, but must alternatively also consider the following reasons: (a) persistently high pulmonary vascular resistance due to initially insufficient lung aeration, (b) low cardiac output due to ineffective chest compressions, or (c) undetected tube occlusion. The approaching team suspected esophageal intubation and removed the endotracheal tube immediately. Colorimetric CO_2_ detectors are useful to indicate correct endotracheal tube placement after lung aeration and consecutive increase in lung perfusion. They have also been used to identify patent airways during non-invasive PPV [25], whereas continuous measurement of end-tidal CO_2_ may be used to guide and optimize chest compressions during CPR [26]. Current resuscitation guidelines recommend “umbilical venous catheterization as the primary method of vascular access during newborn infant resuscitation in the delivery room”, but define intraosseous vascular access as “reasonable alternative” [2]. At our institution the choice of emergency vascular access (umbilical venous catheter or intraosseous access) during neonatal resuscitation is at the treating neonatologist’s individual discretion; in the described case intraosseous access was chosen primarily and could be established on the second attempt, providing us with an effective route for administration of emergency drugs. In a previous study we found that although infrequently required, intraosseous access was successfully used during resuscitation of preterm and term neonates with a moderate overall success rate of 75% [27]. No study so far has directly compared drug administration between the umbilical venous and the intraosseous route; however, Schwindt et al. [28] compared umbilical venous catheterization versus intraosseous access in the simulated environment and found significantly reduced time to successfully establish intraosseous access. Scrivens et al. [29] included 41 neonates in their systematic literature review of intraosseous access, and concluded that it “should be available on neonatal units and considered for early use in neonates where other access routes have failed”.

When considering the crisis resource management, one very fortunate aspect of this unusual resuscitation scenario was the time of day, which happened to be on a weekday at the time of handover. This meant that several neonatologists and one pediatric intensive care specialist were readily available and rapidly involved. These personnel resources allowed for prompt and efficient delivery of resuscitative measures, and likely contributed to the overall excellent outcome of the patient.

## 4. Conclusions

We describe the rare clinical presentation of a neonate with perinatal asphyxia, who required postnatal resuscitation and defibrillation due to VF following epinephrine administration. Based on this case we suggest that medical professionals managing neonatal resuscitation be aware of the possible need for defibrillation, even though it may be very rare. Institutional simulation-based training of neonatal resuscitation should therefore include scenarios involving cardiac arrest due to shockable heart rhythms (i.e., VF and pulseless ventricular tachycardia) to prepare neonatal resuscitation teams to manage such infrequent events adequately. In neonates with evident QTc prolongation epinephrine should be used with caution due to its arrhythmogenic properties, especially the higher recommended dosage of 30 µg/kg [2]. Finally, we consider it appropriate to have a defibrillator, allowing for weight-adapted, gradual titration of the energy level, and appropriately sized pediatric defibrillation pads available in every delivery room.

## Figures and Tables

**Figure 1 children-08-00421-f001:**
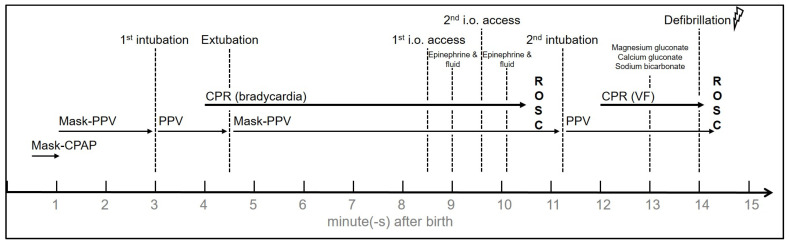
Time course of medical interventions (CPAP: continuous positive airway pressure; CPR: cardio-pulmonary resuscitation; i.o.: intraosseous; PPV: positive pressure ventilation; ROSC: return of spontaneous circulation; VF: ventricular fibrillation).

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
