# Peer review of "Successful Postnatal Cardiopulmonary Resuscitation Due to Defibrillation"

_children, 2021, doi:10.3390/children8050421_

Round 1

Reviewer 1 Report

Dear Authors, I would like to congratulate with you for the case report entitled "Successful postnatal cardiopulmonary resuscitation due to defibrillation" . I have few comments: 1. As you described in the discussion, postnatal VT is very rare and mainly related to long QT syndrome or ALCAPA. Did the patient undergo a complete echocardiographic evaluation to rule out any congenital heart disease? Did the ECG performed at hospital admission show long QT? 2. You calculated QTc both before and during whole body cooling, however hypothermia is known for causing QT interval prolungation. Please, discuss that. 3. Did you check electrolytes?

Reviewer 2 Report

This is a rare and interesting clinical case. The writing of the article is correct. Please take into account the following considerations:

1.- The authors should explain why they chose to put an intraosseous line instead of catheterizing an umbilical vein in this case, when the umbilical line is the one of choice in neonates at birth according to international guidelines. Is it the usual clinical practice in your workplace? Please explain.

2.-  The following bibliographic citations should be mentioned and included when discussing  defibrillation in the neonatal period:

Hirakubo Y, Ichihashi K, Shiraishi H, Momoi MY. Ventricular tachycardia in a neonate with prenatally diagnosed cardiac tumors: a case with tuberous sclerosis. Pediatr Cardiol. 2005 Sep-Oct;26(5):655-7. doi: 10.1007/s00246-004-0714-5. PMID: 16132312.

Sauer CW, Marc-Aurele KL. A Neonate with Susceptibility to Long QT Syndrome Type 6 who Presented with Ventricular Fibrillation and Sudden Unexpected Infant Death. Am J Case Rep. 2016 Jul 28;17:544-8. doi: 10.12659/ajcr.898327. PMID: 27465075; PMCID: PMC4968432.

This bibliographic citation is already included in a previous part of the text

3.- Please note in the text that your group is the author of one of the referenced citations (25). We recommend changing the wording to make this clear. It would be convenient to add some other reference on the use of intraosseous routes in the delivery room and its comparison with the umbilical venous access.

Round 2

Reviewer 1 Report

Dear Authors,

Thank you very much for addressing all the comments raised by the revewers.